# The Behaviour of Sheep around a Natural Waterway and Impact on Water Quality during Winter in New Zealand

**DOI:** 10.3390/ani13091461

**Published:** 2023-04-25

**Authors:** Aloyce Bunyaga, Rene Corner-Thomas, Ina Draganova, Paul Kenyon, Lucy Burkitt

**Affiliations:** 1College of Veterinary Medicine and Biomedical Sciences, Sokoine University of Agriculture, Morogoro P.O. Box 3020, Tanzania; 2School of Agriculture and Environment, Massey University, Private Bag 11222, Palmerston North 4442, New Zealand; r.corner@massey.ac.nz (R.C.-T.); i.draganova@massey.ac.nz (I.D.); p.r.kenyon@massey.ac.nz (P.K.);

**Keywords:** behaviour, ewes, waterway, water quality, winter

## Abstract

**Simple Summary:**

The impacts of extensively managed sheep on the natural environment has received little attention in comparison to beef and dairy cattle in New Zealand. In particular, there is a paucity of information on the interaction of sheep with natural waterways and their impact on water quality. The current study was designed to determine the behaviour of sheep on a hill country paddock, which was transected by a natural waterway, and assessed measures of water quality during winter conditions. The study also investigated sheep behaviour and impact on the water quality of the waterway when they had access to a reticulated water trough. Observations of behaviour showed that sheep spent little time near the waterway compared to other areas of the paddock. In addition, access to a water trough had no effect on ewe time spent grazing, walking, resting, and drinking. Ewes had minimal interaction with the waterway. Hence, under the current conditions of the study, sheep had little impact on the waterway. This may have been due to the high moisture content of pasture during winter; thus, the sheep were not required to interact with the waterway.

**Abstract:**

Access of livestock, such as cattle, to waterways has been shown to be a cause of poor water quality due to pugging damage and excretion entering the water. In New Zealand, regulations require that cattle, deer, and pigs are excluded from accessing waterways, but there are no such requirements for sheep. The current study utilised 24 h video cameras, global positioning system units, and triaxial accelerometers to observe the interaction of Romney ewes (*n* = 40) with a natural waterway. Ewes were either restricted (week 1) or given access to a reticulated water trough (week 2). Proximity data showed that ewes spent more time within 3 m of the waterway when the trough was unrestricted than when restricted (14.1 ± 5.7 and 10.8 ± 5.1 min/ewe/day, respectively; *p* < 0.05). Ewes travelled shorter distances on the steeper areas of paddock than flatter areas. Similarly, ewes showed a spatial preference for the flat and low sloped areas of the paddock. Concentrations of suspended sediment and total phosphorus were higher during access to a reticulated water trough which coincided with the week with more rainy days. Phosphorus and *E. coli* concentrations in the stream water samples were the above recommended Australian and New Zealand Environment and Conservation Council water quality guidelines, especially after rainy days, but did not appear to be directly related to sheep activity. Overall, the results suggest that during winter, ewes interacted very little with the waterway and were thus unlikely to influence the levels of nutrient and pathogens in the waterway.

## 1. Introduction

The farming of livestock on New Zealand’s hill country environment can have negative effects on water quality due to the contamination of waterways with phosphorus (P), nitrogen (N), sediment, microorganisms, and faecal matter [1,2,3,4]. Nitrogen, *Escherichia coli* (*E. coli*), and P contamination can originate from animal urine and faeces, while fertiliser can be a potential source of N and P [5,6,7]. For example, fertiliser and dung as sources of P contribute up to 40% of the total amount, and the rest is from other sources such as pasture-plants and soil components [7]. Fertiliser and manure account for 57% of the total nitrogen contamination in watersheds [8]. High levels of N and P in waterways can lead to excess algal and aquatic macrophyte growth [9,10], which may alter algal community structure and function [11]. In addition, excess N and P can potentially make water unsafe for drinking for both stock and humans [12,13]. 

Sediment that is washed into waterways can also contain significant concentrations of P and *E. coli* [14,15]. *E. coli*, present in animal faeces, serves as an indicator organism for faecal contamination and microbiological impairment of waterways [6]. Increased faecal microbes can pose a health risk to animals, water life, and humans [16,17,18]. Further, excess sediment in waterways can negatively impact the aquatic habitat by limiting penetration of light through the water, decreasing photosynthesis of aquatic ecosystems, and impacting human water use [19,20,21]. Therefore, there is a need to control the deposition of animal excreta into waterways and movement of animals, causing sediment disturbance around these areas.

In May 2020, the New Zealand government released their “Essential Freshwater” package which contained changes to the existing National Policy Statement for Freshwater Management. These polices included new regulations requiring that cattle, pigs, and deer be excluded from waterways more than 1 m wide and located on low slopes [22]. Currently, there is no such requirement for sheep. To date, the behaviour of sheep around natural waterways has received little attention, and there is a need to monitor behaviour to determine if their access to waterways should be restricted. Recently, there has been an increase in the use of digital technologies for animal behaviour studies which include the use of video cameras, tri-axial accelerometers, and global positioning systems (GPS) [23,24,25,26,27,28]. The advantage of these digital technologies include reduced labour and time costs associated with undertaking behavioural measures, continuous observation of animals regardless of the time of the day [29,30], and reduced bias due to observer effects [31,32,33]. 

Little information is currently available on the behaviour of sheep around natural waterways. Sheep prefer natural water to reticulated water for drinking. They may choose to graze close to waterway areas if the pasture is lush [34]. On many New Zealand farms, sheep have access to free water from reticulated water supplied in troughs. Sheep can also obtain water from the pasture they consume; these pastures can contain up to 70% moisture [35]. It has been reported that in winter, when the dry matter content of pasture was less than 30–50%, sheep were able to meet their water requirements from pasture alone and did not need additional free water [35,36]. It is therefore likely that during winter, sheep do not need to drink water. Therefore, natural water sources, such as a stream, are not required to meet their water intake needs. 

The current study firstly examined the behaviour of sheep around a natural stream and secondly studied their impact on water quality in the stream. This study was conducted in the presence and absence of a reticulated water trough to examine the impact this may have on their behaviour. It was hypothesised that sheep would have little interaction with a natural waterway and that these interactions would not be influenced by the access to a reticulated water trough. It was also hypothesised that sheep would not have an impact on the water quality of the natural waterway.

## 2. Materials and Methods

All the procedures in this study were carried out with the approval of the Massey University Animal Ethics Committee (MUEC 19/62). The study was conducted over a 15-day period from 16 August 2019 (D1) to 30 August 2019 (D15) at Massey University’s hill country farm, Tuapaka, located approximately 15 km north-east of Palmerston North, New Zealand (40.3346° S, 175.7316° E), with study paddock located at 40.3345° S, 175.7390° E.

### 2.1. Study Site

The study was conducted in a permanently fenced 1.7 ha paddock with the dimensions of 249.0 m × 249.4 m × 85.0 m × 50.2 m (Figure 1) that contained a discrete natural stream. The stream was classified as sixth-order based on high resolution 1 m LiDAR digital elevation data [37]. The stream was 233 m in length, <1 m wide, and <30 cm deep in base flow conditions (Figure 1). The stream contained culverts at the entrance and exit of the paddock. The watershed area supplying the study paddock between the inflow and outflow was calculated by ArcGIS Pro to be 4.1 ha (Figure 2, watershed 2).

#### 2.1.1. Animals and Study Design

The study utilised mature (3 to 5 years of age) Romney ewes (n = 40) that had been diagnosed bearing a single foetus by trans-abdominal ultrasound at approximately day 90 of gestation. Prior to the study, ewes were managed in an extensive pastoral system and offered 100% ryegrass (*Lolium perenne*) and white clover (*Trifolium repens)* pasture within a rotational grazing system. The ewes in the study were familiar with the paddock as it was part of their normal grazing rotation. Prior to the start of the study, the pasture mass of the paddock was measured to ensure that sufficient dry matter was present to allow ewes to remain at the study site for the entire two-week period. 

During the study period, a crossover design was utilised. Ewes were grazed in the study paddock for one week while being offered access to a reticulated water trough (unrestricted), followed by a second week when the trough was covered resulting in free water only accessible from the stream (restricted). Ewe movement within the paddock and interaction with the waterway was monitored using GPS, triaxial accelerometers, and video surveillance footage. Water samples were collected to measure indicators of water quality. Throughout the study period, ewes were continuously stocked on a predominantly perennial ryegrass pasture (*Lolium perenne*) with masses greater than 1000 kgDM/ha and an average moisture content of 77%. 

#### 2.1.2. Sheep Behavioural Observations

Ewe movement and behaviour around the stream was recorded using fourteen movement activated video surveillance cameras (Moultrie^®^ model MCG-13297, Birmingham, AL, USA, n = 6; TechView^®^ model QC8027, Kaki Bukit, Singapore, n = 4; Bushnell^®^ model 119736, Overland Park, KS, USA, n = 4). Cameras were placed at intervals of 14 to 18 m along the length of the stream (Figure 1). Cameras were triggered by movement of sheep within 15 m of the unit. Footage from the cameras allowed the identification of individual sheep up to a distance of 15 m. Once triggered, cameras were programmed to record for 30 s, followed by a non-recording period of 10 s. If an animal was still moving within the range of the camera after the non-recording period, a further 30 s of footage was recorded. The cameras contained an infrared LED flash that allowed the capture of footage during hours of darkness without visible light being emitted. In order to identify each individual animal, each ewe was marked with large, coloured numbers on their sides using stock spray (Sprayline, Donaghys, Christchurch, New Zealand; Figure 3) and were fitted with a plastic collar labelled with a unique number. 

Ewe behaviours were determined from video recordings using the ethogram below (Table 1). The behaviours included grazing, walking, stationary, drinking (further categorised by whether ewe stood on the bank or in the water), sniff water, walk in the stream, and being out of view. Sheep behaviour was coded using behaviour coding software BORIS (version 7.8.2; [38]). Once the coding process was completed, a total duration for single or grouped observations was extracted.

#### 2.1.3. Global Positioning System (GPS) 

All study ewes (n = 40) were fitted with a collar to which a custom-built GPS unit (DataCarter Ltd., Feilding, New Zealand) weighing ~100 g was attached throughout the study period (Figure 3). GPS monitors were programmed to allow for continuous tracking of satellites and logging of animal position whenever ewes moved ≥5 m or every 60 s if the ewe was stationary. Each GPS unit was powered by a 3.6-volt battery (Tadiran™ lithium Inorganic battery, Kiryat Ekron, Israel) with a life under continuous GPS use of 15 to 25 days. Both the GPS and the battery were enclosed in a moulded plastic weather-proof case. GPS units recorded date and time (GMT), latitude, longitude, horizontal dilution of precision (HDOP), and the number of satellites detected. Distance was calculated between two points specified by latitude/longitude (in numeric [decimal] degrees) in excel VBA that was run with macros derived from the Vincenty inverse formula for ellipsoids [39].

#### 2.1.4. Triaxial Accelerometers

All study ewes (n = 40) were also fitted with a triaxial accelerometer with the dimensions of 4.6 cm × 3.3 cm × 1.5 cm. That weighed 19 g (wGT3X-BT Actigraph, Pensacola, FL, USA) and was attached to the collar for the study period. The accelerometers contained Bluetooth^®^ technology (N. semiconductor, Trondheim, Norway) and were set to be “beacons”. Beacons sent signals containing their ID number at 10 s intervals to other accelerometers (receivers), indicating the proximity or location of a device. The receiver accelerometers were attached to the posts to which the video cameras were attached and recorded the proximity of any beacon devices once per minute. The distance between beacons and receivers was estimated using the received signal strength indicator value (RSSI) and transmitted power of the beacon. In the current study, three meters on either side of the stream zone corresponded to RSSI values below −56 dB (in Excel IF (A2 <= −56,1,0)). The accelerometers were initialised using proprietary software (ActiLife software, version V6.13.4, ActiGraph LLC, Pensacola, FL, USA), during which the device was identified with the sheep number or camera location, the start date, and time. The sample rate of 30 Hz was set. The accelerometers were programmed to continue to collect data until the battery was depleted. 

#### 2.1.5. Ewe Measures

At D1 and D15, ewes were weighed (Tru-Test weigh scale, Auckland NZ), and their body condition scored by a single experienced technician [scale 1–5; [40]]. On each occasion, ewes were weighed within an hour of being removed from the pasture. 

#### 2.1.6. Pasture Measures

The moisture content of the pasture was determined on D1, D10, and D15. At approximately 1 pm on each day, 6 grab pasture samples of approximately 50 g were collected from a height of between 5 and 15 cm above the ground to simulate sheep grazing. Samples were randomly collected across the paddock by hand plucking. Samples were oven-dried at 80 °C for 48 h in order to determine dry matter yield and moisture content. 

Moisture content Equation (1)
(1)$$\left(\frac{Fresh\;weight-Dry\;weight}{Fresh\;weight}\right)\times 100$$

Pasture mass was estimated on D1, D10, and D15 using a manual folding plate meter (Jenquip, Fielding, New Zealand). One hundred readings were taken across the paddock at approximately 2 m intervals, and the average mass for the paddock was recorded. Pasture mass was calculated from the equation below [41]

Pasture mass Equation (2)
(2)$$Pasture\;mass=\left(Final\;reading-Initial\;reading\right)\times 158+200$$

#### 2.1.7. Weather Data

Hourly and daily data were downloaded from Tuapaka weather station 2 (EnviroMonitor station, Davis Instruments, Davis, CA, USA), located 800 m from the study site. Data included rain, relative humidity, air temperature, solar radiation, and wind speed.

#### 2.1.8. Water Measurements

Water flow rate was measured, and samples were collected hourly for 8 h (from 8 am to 3 pm) on D5, D6, D12, and D13 (Table 2). The water flow rate of the stream was measured as water exited each of the two culverts: one at the entry (inflow) and one at the exit (outflow) of the paddock (Figure 1). The flow rate was manually calibrated using a 30-litre flexible bucket, stopwatch, and a graduated jug, as per the method described by Abd Ghani [42]. Each hour, the process was repeated three to five times and the average calculated. The equation to calculate the bucket flow rate was

Water flow rate Equation (3)
(3)$$Flow\;rate\;(L/s)=\frac{Volume\;of\;water\;in\;the\;bucket\;(L)}{Time\;to\;fill\;(s)}$$
animals-13-01461-t002_Table 2Table 2Timeline of the study showing the calendar date, study day, and water sample collection.Date13-Aug14-Aug15-Aug16-Aug17-Aug18-Aug19-Aug20-Aug21-Aug22-Aug23-Aug24-Aug25-Aug26-Aug27-Aug28-Aug29-AugStudy DayD-3D-2D-1D1D2D3D4D5D6D7D8D9D10D11D12D13D14Water Sampling-------√√-----√√-

Water samples were collected into 1 L plastic bottles from the inflow and outflow of the paddock to determine the concentration of suspended sediments (SS), total phosphorus (P), nitrate-N, ammonium-N, and *E. coli*. Water samples were collected synchronously on the hour at both culverts. At the same intervals, additional samples to determine *E. coli* concentration were collected into 100 mL sterile microbiology bottles (Eurofins, Wellington, New Zealand) using a sterile technique to avoid sample contamination. 

All water samples were stored in a cool box with ice until they were returned to the laboratory at the conclusion of the sampling day. *E. coli* samples were then couriered to the analytical laboratory on the same day. Each 1 L sample was subsampled, with one subsample being filtered to <0.45 µm for nitrate-N. Ammonium-N analysis and the second subsample were left unfiltered for total N and P analysis. Subsamples were then frozen (−20 °C) for subsequent analysis. The remaining ~900 mL sample was refrigerated at 4 °C for subsequent SS analysis. 

#### 2.1.9. Nitrate-N, Ammonium-N and Total P Concentrations

The nitrate-N and ammonium-N concentration of the water samples was determined using a colorimetric autoanalyzer (Pulse international ltd, Saskatoon, SK, Canada) method using red azo and phenol Prussian blue dyes for nitrate-N and ammonium-N, respectively [43]. Quality control was assessed by analysing solutions of known concentrations of 12, 0, 0.25, 0.5, 1, 2, 4, 8, 12, and 0 ppm in sequential order. Two blank (0 ppm) and a standard solution (8 ppm) were also analysed every ten samples to monitor the accuracy of the results measured. Detection limits for both nitrate-N and ammonium-N using this method was 0.25 mg/L. Total P (TP) concentrations were determined by digestion and automated ascorbic acid colorimetry (APHA 4500-PH method) using a flow injection analyser [44]. 

#### 2.1.10. Suspended Sediment Analysis 

The concentration of SS was determined using the gravimetric analysis following the standard procedure from the American Public Health Association (2005). A thoroughly homogenised sample was passed through a previously rinsed, dried, and weighed filter paper (Whatman 100 mm GF/C). The filter paper (with sediment) was then dried at 105 °C for >8 h and re-weighed. The empty sample bottle was also weighed. The suspended sediment concentration was calculated using the equation below.

Suspended sediment Equation (4)
(4)$$mg\;total\;suspended\;sediment/L=\frac{\left(A-B\right)\times 1000}{Sample\;volume\;in\;mL}$$
where *A* is the weight of filter paper + dried residue in mg, and *B* is the weight of filter paper in mg.

#### 2.1.11. *E-coli* Analysis

*E. coli* concentrations were determined within 24 h of collection by Eurofins laboratory (Wellington, NZ) using a membrane filtration procedure (Standard Method APHA 9222G; [44]) onto nutrient agar containing 4-methylumbelliferyl beta-D-glucuronide. *E. coli* results were reported as colony forming units (cfu) per 100 mL of sample.

#### 2.1.12. Statistical Analysis

GPS data were analysed using ArcGIS mapping tools (ArcGIS Pro 2.2.4, 2018). Distance travelled by sheep data were cleaned and processed using in-house macros in Microsoft Excel. Distance travelled per sheep was analysed by time of day categories which included early morning (0600 to 0800), day (0900 to 1600), evening (1700 to 1900), and night (2000 to 0500). 

An optimised hot spot analysis (z-score) was conducted using ArcGIS to identify statistically significant spatial clustering of ewe GPS location fixes. A hotspot was defined as an area of higher concentration of ewe locations compared to the expected number given a random distribution of ewes. A cold spot was defined as an area that had a lower concentration of ewe locations compared to the expected number given a random distribution of ewes. The analysis of ewe interaction with waterway was based on an area defined as the ‘stream zone’ which represented the stream and 3 m either side.

Prior to analysis, ewe behaviour, and water quality data were checked for normality using the Kolmogorov–Smirnov and Shapiro–Wilk test, and the homogeneity of variances was studied using Levene’s test and Tukey transformation (Tukey’s Ladder of Powers transformation), where appropriate [45]. Behavioural data, including grazing, drinking, and walking analyses, were performed using R 3.6.0 (2019-04-26; R Core Team (2019)). The impact of water trough restriction on ewe behaviour (e.g., % grazing, drinking) was analysed using a 2-factor analysis of variance (ANOVA). A linear regression was also used to determine if any ewe behaviours were associated with time of the day or environmental temperature.

Concentrations of *E. coli*, N, and SS concentrations were analysed using parametric and TP using non-parametric methods in R 3.6.0. The load of nitrate-N, SS, *E. coli*, and TP in inflow and outflow water samples was calculated as the product of their concentration and stream flowrate (Equation (5)). One way ANOVA analyses were followed by post hoc test when significant (Tukey test *p* < 0.05); otherwise, non-parametric ANOVA models (Kruskal–Wallis) were used to assess differences between mean ranks.

Load Equation (5)
(5)Load (mg/s)=Concentration (mg/L)×flowrate (L/s)

## 3. Results

### 3.1. Weather

Weather data were retrieved from three days prior to the start of the study (D-1 to D-3) to the completion of the study period (Figure 4). Rainfall occurred on 7 days of the 16-day study period, with rainfall volumes ranging from 0.2 to 27 mm/day. Maximum daily temperatures ranged from 7.2 to 13.1 °C, and the minimum daily temperature ranged from 1.7 to 8.4 °C. Relative humidity ranged from 64% to 90%. 

### 3.2. Stream Flowrate

Stream flowrates at the inflow and outflow monitoring sites ranged from 6.99 to 26.78 L/s but fluctuated slightly during each sampling day (Figure 5a,b). Flowrate did not differ between the water trough restricted and non-restricted periods (*p* = 0.98). Correlations between flowrate and weather parameters showed a negative correlation between flowrate and daily average temperature (r = 0.4 *p* = 0.002).

### 3.3. Water Quality

Water quality analyses showed that concentrations of SS and TP differed (*p* < 0.05) between the periods when the water trough restricted and unrestricted; however, there was no difference (*p* > 0.05) in *E. coli* or nitrate-N (Table 3). Ammonium-N was below the detection limit of the analytical method throughout the study period.

The nitrate-N load measured at the outflow sampling site was higher than at the inflow site at D5 but lower at D13 (*p* < 0.05; Figure 6a). Suspended sediment loads were higher at the outflow than inflow site on D5 and D6 during the restricted period but did not differ during the unrestricted period (*p* < 0.05; Figure 6b). *E. coli* loads were higher in the outflow than inflow site at D12 (*p* < 0.05; Figure 6c) but did not differ on D5, D6, or D13 (*p* > 0.05). The total P load did not differ between sampling sites at any time during the study (*p* > 0.05; Figure 6d). 

Across the four sampling days, the load of nitrate-N, TP and SS did not vary by hour of the day (*p* > 0.05). At D13, however, *E. coli* load was higher at 10:00 am than the rest of day (*p* = 0.015).

### 3.4. Animal Density and Spatial Distribution

Ewe GPS locations across the study site showed that there were more fixes in the northeast than southwest areas of the paddock (Figure 7C,D). Based on the slope profile shown in Figure 7B, these areas coincided with the flat areas (0–3°) of the paddock. During the period of water trough restriction (Figure 7C), there were fewer areas with aggregated GPS locations (focal areas) identified than when the trough was not restricted (Figure 7D). There were, however, more ewe locations recorded near the trough when access to the trough was restricted compared to unrestricted (Figure 7C,D). Regardless of the period of the study, there were proportionally fewer ewe GPS locations recorded near the stream (n = 6829) than the rest of the paddock area (n = 712,400). 

During the period of water trough restriction, there were two significant hot spots (statistically significant spatial clustering of locations) identified within the paddock (*p* < 0.05; Figure 8A). When the trough was unrestricted, eight smaller areas were detected (Figure 8B). When the stream zone was analysed independently of the rest of the paddock, six cold spots and five hot spots were identified during the unrestricted trough access period (*p* < 0.05; Figure 8D), whereas in the restricted period, the stream zone showed only hot spot areas around each culvert (Figure 8C). 

The stream zone made up 9% of the entire paddock area. However, 0.7 and 1.0% of all ewe GPS location fixes were recorded within this area during the restricted and unrestricted water trough access periods, respectively (Table 4). In the stream zone, 40% of all ewe GPS location fixes during the unrestricted period were recorded in the three hot spot areas (Figure 8D), whereas the same areas contained 24% of all GPS location fixes in the restricted period (Figure 8C). 

### 3.5. Effect of Slope

Within the paddock, areas with a slope of less than 15° contained more than half of the sheep location fixes recorded during the entire study period (Table 4). Fewer than 2% of GPS location fixes were detected in areas with a slope of greater than 35° (very steep; Table 4). 

### 3.6. Ewe Distance Travelled 

The mean distance travelled per ewe per hour (m/h) varied throughout the day (Figure 9). Peaks were observed between 0400 and 0500, between 1400 and 1700, and between 2000 and 2100 h. Ewes travelled greater distances when the trough access was restricted (149 ± 13.8 m/h) compared to unrestricted (114 ± 8.8 m/h; *p* = 0.046; Figure 9). Over the entire study period, ewes travelled further (*p* < 0.05) during the night (148 ± 18.3 m/h) and in the evening (156 ± 6.8 m/h) than either in the early morning (131 ± 19.6 m/h) or day (100 ± 20.1 m/h).

The mean distance ewes travelled differed across paddock slope classes (*p* < 0.05). In general, as slope increased, ewes travelled shorter distances; however, similar distances were travelled on strong rolling, moderately steep, and steep slopes in the unrestricted period (Table 5). During the period of water trough restriction, distances travelled were similar between the flat and strongly rolling and between rolling and moderately steep areas (*p* > 0.05; Table 5) but differed across other slope classes. 

### 3.7. Time Spent in the Stream Zones 

Over the entire study period, proximity data showed that the time ewes spent within the stream zone (3 m either side of the stream) differed between camera locations (*p* = 0.005). Ewes spent the most time near camera 3 (19.8 ± 6.7 min/ewe/day) and camera 11 (22.2 ± 2.3 min/ewe/day). When access to the trough was unrestricted, ewes spent more time (*p* < 0.05) within 3 m of any camera location (14.1 ± 5.7 min/ewe/day) than when access was restricted (10.8 ± 5.1 min/ewe/day: *p* = 0.018). 

The duration that ewes were within 3 m of a camera location differed (*p* < 0.05) between time-of-day classes. Whereby, ewes were in proximity of any camera position longer in the early morning (10.8 ± 5.4 min/ewe/day) than during daylight hours (8.6 ± 4.4 min/ewe/day; *p* = 0.024). In addition, the median duration near any camera was greater in the evening (12.5 ± 7.2 min/ewe/day) than day (*p* = 0.004) and between evening and night (9.1 ± 5.4; *p* = 0.001). 

A linear regression showed that daily distance travelled, average air temperature, and relative humidity accounted for 76% of the variance in the time ewes spent within 3 m of a camera. For every additional degree of temperature, ewes spent 1.2 min more within 3 m of the stream. Similarly, for every one percent increase in relative humidity, ewes spent 0.3 min more within 3 m of the cameras. 

### 3.8. Behaviour in the Stream Zone

Video footage showed that during the entire study period when ewes were in the stream zone, they spent 68% of their time grazing (n = 96), 15.9% stationary (n = 52), 11.2% walking (n = 53), and 2.2% (n = 5) drinking from the stream. Ewes were observed to sniff water on five occasions (0.5%). On one occasion, an ewe was observed to walk in the stream (0.1%). The duration spent undertaking grazing, stationary, and walking behaviours did not differ (*p* > 0.05) between periods when the water trough was restricted or unrestricted. Video footage from both treatment periods showed that the behavioural events observed at hotspot two where there was no culvert (Figure 8D) included crossing the stream (12.5%), while the remaining 87.5% was grazing. 

The duration ewes were observed to be stationary showed a negative relationship with solar radiation (r = −0.8, *p* < 0.05). For each additional one MJ/mÂ^2^ of solar radiation, there was an associated 0.48 s decrease in average time spent stationary. The average duration ewes spent undertaking all other behaviours during the entire study period was not affected (*p* > 0.05) by temperature, relative humidity, or wind speed.

### 3.9. Behavioural Events at the Outflow Culvert 

During the study period, there were 185 behavioural events recorded at the outflow culvert (camera 1) which included sheep crossing the stream over the culvert (n = 145, 78.4%), crossing to graze (n = 17, 9.2%), grazing near the stream (n = 22, 11.9%), and resting (n = 1, 0.5%; Figure 10). All the ewes (n = 40) were observed to utilise the outflow culvert at least once during the study. Most of the ewes (n = 32) utilised the culvert more than once. Four ewes were observed to utilise the culvert ≥10 times during the study. Forty percent (n = 74) of the behavioural events were recorded during the period of unrestricted water trough access compared to 60% (n = 111) during the restricted period (*p* = 0.87). Of the behaviours observed, 68% was recorded during the day (9 a.m. to 4 p.m.) compared with 25% in the morning (6 to 8 a.m.) and 7% the evening (5 to 7 p.m.); time of day only tended to affect the frequency of culvert use (*p* = 0.09).

## 4. Discussion

The aim of the current study was to examine the behaviour of sheep in and around a natural waterway when access to a water trough was unrestricted or restricted. The potential impacts of sheep on water quality were also assessed. It was hypothesised that the behaviour of the ewes around the waterway would not differ when given access to a trough or access was restricted. It was also hypothesised that the interaction of sheep with the natural waterway would not influence the level of nutrient and pathogens in the waterway. 

The spatial distribution of sheep within the paddock was influenced by features such as slope and the location of culverts. Ewes showed a spatial preference for the flat to low sloped areas of the paddock, with more than 70% of GPS location fixes recorded in areas with a slope of less than 15°. This finding is in agreement with a number of studies in sheep which have reported a preference to graze slopes <30° [46,47,48]. Low slopes (1–12°) have been reported to have a greater accumulation of herbage, compared to steeper slope classes [49]. Thus, the spatial preference observed in the present study may have been due to greater pasture availability in these flatter areas [50]. Pasture masses were measured in the current study; however, different slope classes were not recorded separately. 

The distance travelled by ewes in the current study was greater during the period when access to the water trough was restricted than when the trough was accessible. There are limited studies in temperate climates such as NZ. In the tropics, sheep have been reported to be motivated to travel longer distances, especially when feeding or searching for water [51,52]. Therefore, the longer distances travelled during the restricted period of the current study may have been the result of ewes spending more time in search of lush pasture. The 40 m per hour difference in distance travelled between the restricted and non-restricted period was small and unlikely to be of biological significance.

In the current study, ewes spent 0.8% of the day (12 min/day) within the stream zone which constituted 9% of the paddock area. While in the stream zone, ewes were observed to drink and walk in the stream for 0.1% of the day, and only 1% of the ewe GPS locations were recorded in the stream zone. This suggests that in winter, ewes had minimal interaction with the waterway during the study period. Only using Romney ewes was a limitation of the study.

In the stream zone of the study paddock, three GPS hot spot areas were identified where location fixes were denser than the average. These hotspots were located near the inflow and outflow culverts with an additional area in the middle of the stream zone (Figure 8D). Ewes crossing the stream using a culvert is of minimal concern in terms of water quality, as there is no interaction between the animals and the water. The hot spots identified in the stream zone were located in the flattest areas based on slope map. The flattest area of the stream zone represented 3.7% of the total stream zone area but contained 44% of all GPS locations. At hot spot location two, the predominant sheep behaviour observed was grazing (87.5%), which suggests that the pasture in this area was attractive. 

All 40 study ewes accessed the stream zone to graze at some time during the study period, but few (n = 5) were observed to drink from the waterway. Oluju [34] suggested that sheep preferred to graze around waterways to access green pasture within the stream zone. There is little published data, however, on the drinking behaviour of sheep in a temperate environment, and to date, none have focused on drinking from a stream. It is likely that in the current study the ewes had satisfied their water needs through the consumption of pasture. Brown and Lynch [35] and McFarlane and Howard [36] suggest that if the moisture content of the pasture is above 50%, sheep do not need to drink free water [35,36].

In the current study, when ewes were in the stream zone, they spent 68% of their time grazing, 15.9% stationary, and 2.2% drinking. These percentages were similar to Schlecht [52], who reported that 60% of the day was spent grazing and 12 to 20% was spent stationary. Al-Ramamneh [53] and Filipčík [54] reported a shorter time spent drinking than in the current study (0.1% to 0.3%). In their study, Al-Ramamneh [53] observed penned sheep with video cameras that captured the entire area. The difference in drinking percentage between Al-Ramamneh [53] and the current study, therefore, was likely due footage from the current study being recorded only within the stream zone and, thus, did not capture ewe behaviours in other areas of the paddock. Filipčík [54] used human observers to record sheep behaviours every five minutes between 7:15 a.m. and 4:15 p.m., which may have resulted in drinking events being missed. 

Ewes spent on average 3.3 min more per day within 3 m of stream during the period when access to the water trough was unrestricted than when access was restricted. This is biologically of little importance. When within the stream zone, video footage showed that grazing was the dominant behaviour. Previous research suggests that ewe grazing is focused near artificial water-points, and their position can influence sheep grazing activity [55]. In the current study, sheep were observed to spend longer periods within 3 m of camera locations (stream zone) in the early morning and evening than the day and night. This pattern of time spent near the stream was in agreement with other grazing patterns [54,56], where grazing activity was greater in the morning and evening. 

In the current study, both ambient temperature and relative humidity appeared to increase the time ewes spent in the stream zone. It is possible that ewes spent less time within 3 m of the stream zone during the middle of the day, as higher temperatures have been shown to decrease the proportion of time spent feeding. Malpura cross and Polwarth ewes in climatic chambers (18 °C–45 °C) showed a similar pattern (i.e., spent less time at midday) to the current study [57,58]. The decrease in feeding activity may be a result of voluntary adaptive depression of metabolic rate associated with reduced appetite in heat-stressed animals [59]. All the drinking events in the current study occurred when the ambient temperature was greater than 20 °C, which supports the impact of temperature.

Video footage in the current study showed that ewes avoided walking in the water. It also showed that those that did cross jumped over it or crossed using the culverts. This finding was similar to Askey-Doran [60] and Dymond et al. [61], who reported that sheep have reduced affinity to water and show an aversion to standing in water. The tip of their hoof has small contact area with the ground, which may allow the penetration of a film of water and possibly moss, mud, and lichen [62]. In addition, sheep have reduced blood flow to the skin of the lower legs when penetrated by or exposed to cold water [63]. It is possible that this blood flow is connected with drinking behaviour as skin blood flow is adrenergically-mediated due to thermal influences, which may elicit a fear reaction of sheep to water [64]. Water temperature was not measured and is a limitation of the study.

When sheep were given access to or were restricted from accessing the water trough, there was no effect on the proportion of time they were recorded to undertake any of the behaviours observed in the current study. This is perhaps not surprising as they had ad libitum access to pasture which contained 77% moisture, suggesting that sheep could have satisfied their water needs from the ingestion of pasture. This finding is in agreement with a report of desert bighorn sheep in the USA that reported the removal of water catchments did not result in changes in diet, foraging area selection, home-range size, movement rates, mortality, or productivity in ambient temperatures similar to the present study [65].

The higher nitrate-N concentrations measured at the outflow site on D5 and 14 suggested that the catchment area (i.e., the paddock) between the monitoring sites contributed this nitrate-N to the stream, possibly as a result of nitrate-N being leached from the soil to water. Rainfall on day three likely generated both subsurface and surface runoff of N. However, nitrate-N concentrations did not differ between periods of water trough access or restriction. Although concentrations are of interest and mean nitrate-N concentrations measured in the current study (0.82 mg/L) were higher than the 0.226 mg/L reported by Cooper [66] and lower than those reported by McColl and Gibson [67] (1.8 mg/L), given the influence of flow rate on concentration, it important that we also examine loads. Nitrate-N loads were not influenced by periods of water trough access or restriction in the current study, as stream flow rate did not differ between treatments. 

Outflow SS loads were higher on days five and six, when access to the water trough was restricted (Figure 6b), but this may have also been the result of rainfall on day three generating surface runoff into the stream. The hotspot analysis also suggested that SS loads were higher when there was greater spatial clustering of ewes during that period (Figure 8D). Physical disturbance of soil due to trampling can increase the mobilization of sediment [68]. Total P loads followed a similar trend to the SS results, although there was no significant effect of water trough access or restriction in this case. The TP concentration measured in the current study was greater than the Australia and New Zealand Environment and Conservation Council guidelines [69], which suggest that TP should be less than 0.035 mg/L in upland rivers. 

*E. coli* concentrations were higher on day 12 than days 5, 6, or 13. This increase in *E. coli* may have been due to water runoff from a rainfall event on day 12. It is possible that the rainfall washed sheep faeces into the stream, thus resulting in a higher *E. coli* load at the outflow sampling point [70]. In the current study, *E. coli* load was positively correlated with relative humidity and negatively with solar radiation. These findings were consistent with solar radiation reducing the *E. coli* population [71,72]. The New Zealand Ministry for the Environment [22] water quality guidelines for freshwater recreation require *E. coli* concentration to be <130 MPN/100 mL. In the current study, 25% of the samples collected exceeded this concentration. 

## 5. Conclusions 

The current study found that in winter, there was little evidence that sheep interacted with the natural waterway, and there was little impact on nitrate-N or SS. Total P and *E. coli*, however, were recorded to be above suggested thresholds on some days. However, these appeared to be unrelated to the presence of sheep in the paddock, but were instead related to rain events. Ewe behaviour and the degree of interaction of sheep with the waterway were not influenced by the availability of reticulated water from a trough in winter. There was a clear indication that the study ewes showed a spatial preference for flat to low sloped areas of the paddock. Further, long-term studies are required to verify these results. They are particularly required to confirm that the elevated total P and *E. coli* were due to rain events per se and not directly due to sheep interaction with waterways. Other periods of the year also need evaluation.

## Figures and Tables

**Figure 1 animals-13-01461-f001:**
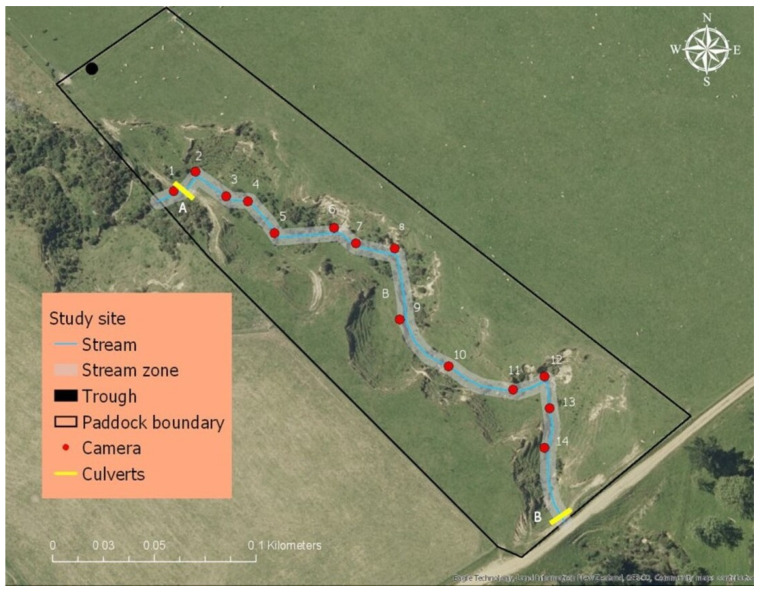
Map of the study site showing the stream (blue line), stream zone (grey shading), and the position of the trough (black dot), culverts (yellow bars (letters A & B)), and cameras (red dots).

**Figure 2 animals-13-01461-f002:**
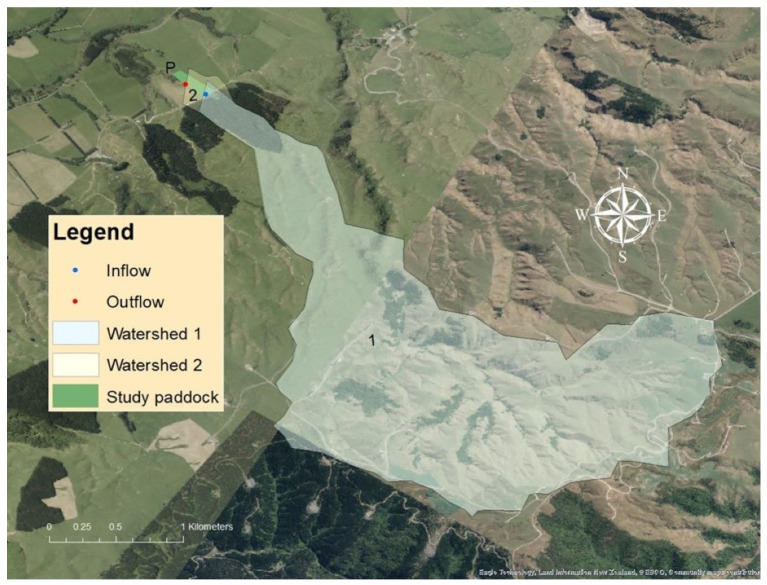
Map of watershed 1 and 2 that supplied the study paddock stream (P). Red and blue dots indicate locations of the outflow and inflow sampling sites, respectively.

**Figure 3 animals-13-01461-f003:**
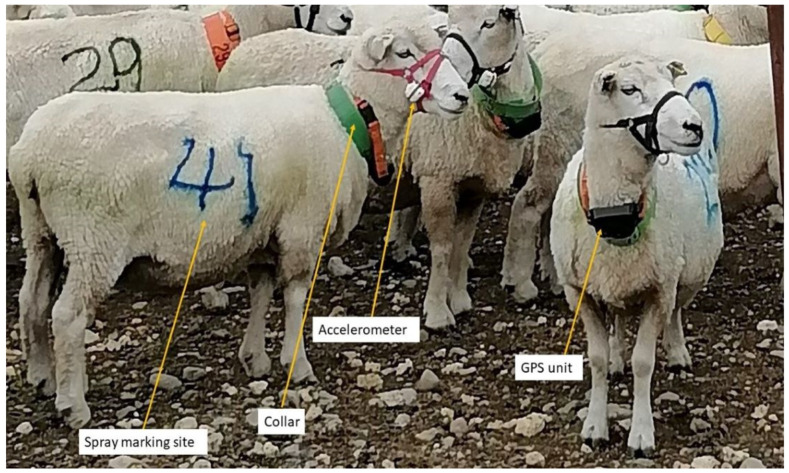
A photo of sheep with spray marks wearing accelerometers, GPS, and collars.

**Figure 4 animals-13-01461-f004:**
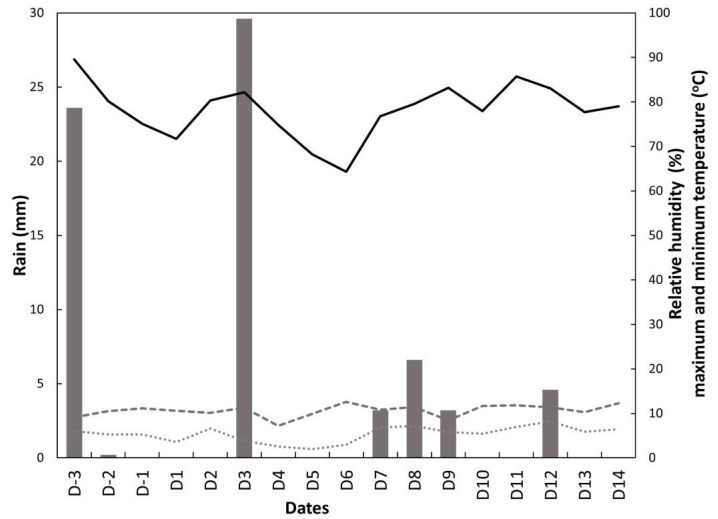
Daily mean rainfall (mm, bar), relative humidity (%, ”______”), Minimum temperature (°C, ”………..”) and maximum temperature (°C, ”--------”) during the study period. D-3 to D14 indicate number of days relative to the start of the study (D1;16 August 2019).

**Figure 5 animals-13-01461-f005:**
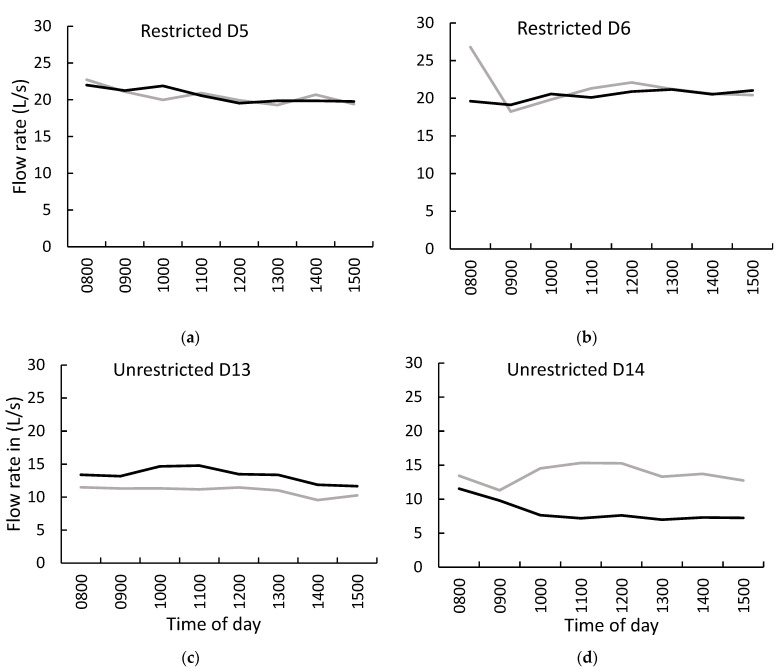
Hourly mean stream flowrate (L/s) at the inflow (grey line) and outflow (black line) monitoring sites by time of day (0800 to 1500 h) during the period of restricted access to water trough on D5 and D6 ((**a**,**b**) in upper panels) and the period of non-restricted access on D13 and D14 ((**c**,**d**) in lower panels).

**Figure 6 animals-13-01461-f006:**
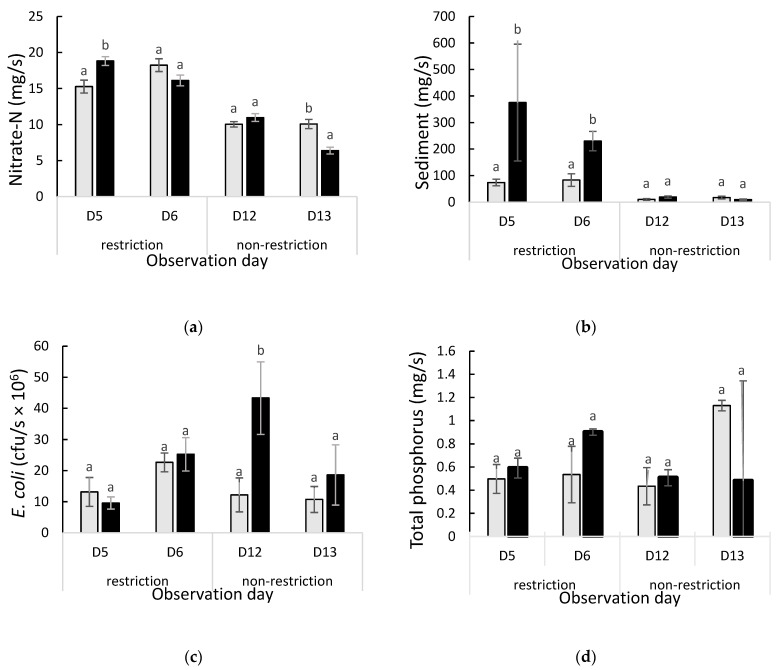
Mean (± SEM) nitrate-N (mg/s; Panel (**a**)), suspended sediment (mg/s; Panel (**b**)), *E. coli* (cfu/s × 10^6^; Panel (**c**)), and total phosphorus loads (mg/s; Panel (**d**)) measured in water samples collected at the inflow (grey bars) and outflow (black bars) sampling sites on study days 5 and 6 (D5 and D6; restricted access to trough) and days 12 and 13 (D12 and 13; unrestricted access to trough). Within each day, bars with different letters were significantly different (*p* < 0.05).

**Figure 7 animals-13-01461-f007:**
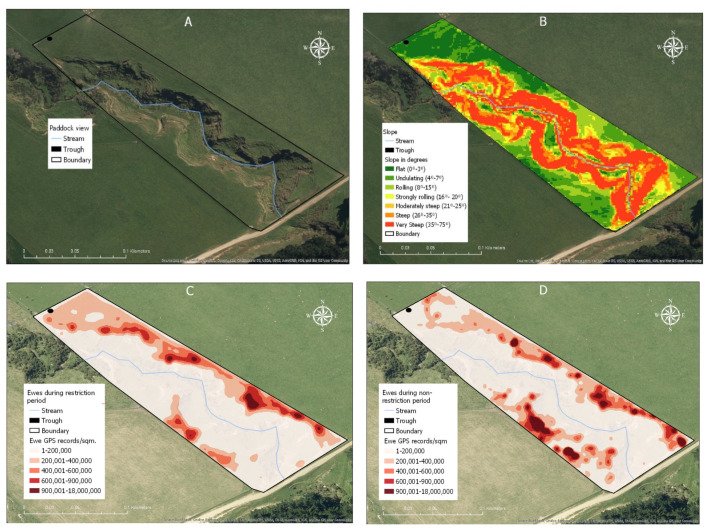
Maps showing a satellite image of the study site (**A**), slope category (**B**), and paddock features including the stream (blue line), water trough (black circle), and paddock boundary (black line); the spatial distribution (magnitude per unit area) of sheep during the period the water-trough was restricted (**C**) or unrestricted (**D**) using kernel smoothing. White areas represent low ewe density (1 to 200,000 GPS locations/m^2^) and red areas represent high density of recorded locations (900,001 to 18,000,000 locations/m^2^).

**Figure 8 animals-13-01461-f008:**
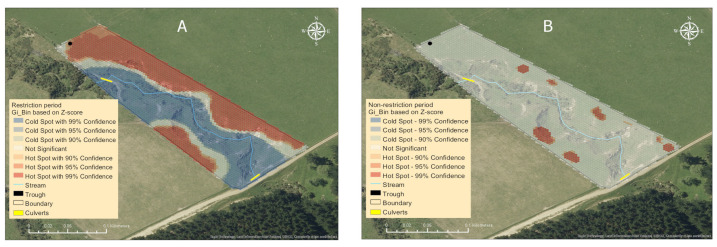
Maps showing the spatial distribution (magnitude per unit area) of ewes within the study paddock (**A**,**B**) and the stream zone (within 3 m of the stream; **C**,**D**) during the period the water trough was restricted (panel **A**,**C**) or unrestricted (panel **B**,**D**) using optimised hot spot analysis. The blue areas represent low ewe density (cold spot) and red areas high ewe density (hot spot (HS)). Hotspots indicate statistically significant (*p* < 0.05) spatial clusters of high values (larger positive z-score), cold spots indicate statistically significant (*p* < 0.05) spatial clusters of low values (smaller negative z-score), and white indicates random distribution with no spatial clustering.

**Figure 9 animals-13-01461-f009:**
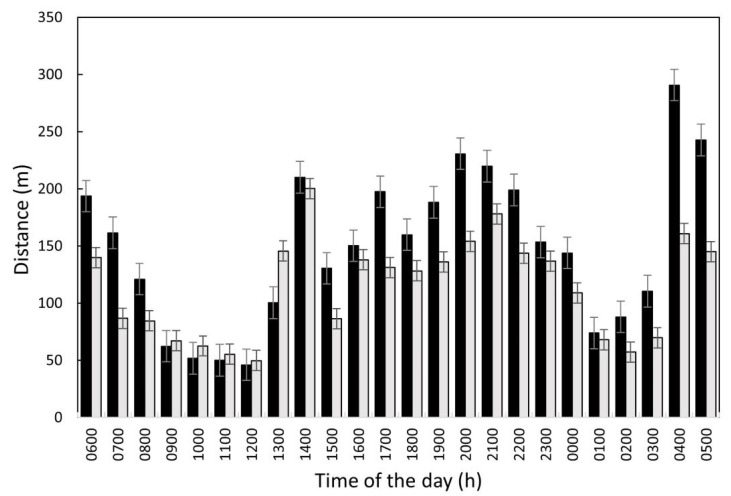
Hourly distance travelled per ewe by hour of the day (m; mean ± SEM) during the period when access to the water trough was restricted (black bars) or unrestricted (grey bars).

**Figure 10 animals-13-01461-f010:**
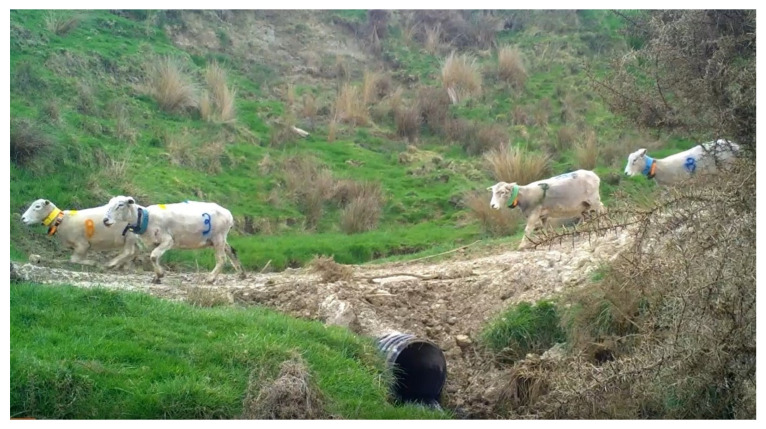
Image captured by video camera number one showing ewes (ID; orange 8, blue 3, green 7 and blue 8) utilising the outflow culvert for crossing.

**Table 1 animals-13-01461-t001:** Ethogram showing the description of ewe behaviours.

Behaviour	Description
Stationary	Ewe was inactive either standing or sitting. Includes sheep that were ruminating (regurgitation; re-chewing; and re-swallowing) or scratching. Standing was defined as all being on four feet, on the ground, and with no locomotion. Sitting was defined as at least 50% of side in contact with the ground and not being supported by all 4 feet.
Grazing	Sheep harvesting vegetation from the ground. Could be standing still or walking with muzzle close to the grass (i.e., head is below shoulders).
Walking	Moving from one point to another did not include walking while grazing.
Drink	Animal consumed water from the stream or trough for more than 5 s.
Sniff	Moved muzzle towards water and inhaled but did not drink.
Walk in stream	Sheep stepped in the stream without drinking water.
Other	Sheep performed any other behaviours, such as playing or fighting.
Out of view	Sheep moved out of range of the video camera

**Table 3 animals-13-01461-t003:** Arithmetic mean (±SEM) and Tukey transformed mean (SEM) of concentration of *E. coli* (cfu/100 mL), nitrate-N (mg/L), suspended sediment (mg/L), and the median (IQR) of total P (mg/L) during periods of restricted and unrestricted access to the water trough.

Parameter	Treatment	N	Mean ± SEM	Transformed Mean ± SEM	Median (IQR)	Treatment *p*-Value
E. *coli* (cfu/100 mL)	Restricted	32	84.72 ± 10.68	−0.2 ± 0.01		0.139
	Unrestricted	32	188.50 ± 43.20	−0.2 ± 0.01		
Nitrate-N (mg/L)	Restricted	32	0.71 ± 0.02	0.5 ± 0.03		0.528
	Unrestricted	32	0.66 ± 0.02	0.4 ± 0.02		
Suspended sediment (mg/L)	Restricted	32	15.08 ± 6.30	−0.8 ± 0.02		0.001
	Unrestricted	32	17.96 ± 16.70	−1.0 ± 0.02		
Total phosphorus (mg/L)	Restricted	32	0.03		0.03 (0.02–0.03)	0.018
	Unrestricted	32	0.07		0.03 (0.03–0.04)	
Ammonium-N (mg/L)	Restricted	32	ND			
	Unrestricted	32	ND			

Restricted = Restricted access to water trough, Unrestricted = Unrestricted access to water trough. ND = Concentration was below the detection limit of 0.25 mg/L.

**Table 4 animals-13-01461-t004:** The mean number (±SE) and percentage (%) of sheep GPS location fixes in each paddock slope class during the period when access to the water trough was restricted or unrestricted.

	GPS Location Fixes (±SE)			
	n		%	
Slope Class (Degrees)	Unrestricted	Restricted	Unrestricted	Restricted
Flat (0–3)	121,397 ± 602 ^e^	120,670 ± 201 ^f^	30.0	38.4
Undulating (4–7)	95,544 ± 757 ^d^	63,812 ± 118 ^e^	23.6	20.3
Rolling (8–15)	62,027 ± 330 ^c^	45,758 ± 93 ^d^	15.3	14.6
Strong rolling (16–20)	27,583 ± 95 ^b^	24,972 ± 55 ^bc^	6.8	8.0
Moderately steep (21–25)	35,730 ± 460 ^b^	31,619 ± 91 ^c^	8.8	10.1
Steep (26–35)	58,879 ± 896 ^b^	20,967 ± 57 ^b^	14.6	6.7
Very steep (35–75)	3393 ± 16 ^a^	6112 ± 86 ^a^	0.8	1.9

^a–f^ Within a treatment group (column), means with different letters were significantly different (*p* < 0.05). Restricted = Ewe access to water trough was restricted. Non-restricted = Ewe access to water trough was not restricted.

**Table 5 animals-13-01461-t005:** Hourly distance travelled by sheep (mean ± SE) for each paddock slope class during the period when access to the water trough was restricted or unrestricted.

	Distance Travelled (m/h)	
Slope Class (Degrees)	Restricted	Unrestricted
Flat (0–3)	69.4 ± 1.36 ^e^	59.6 ± 2.70 ^d^
Undulating (4–7)	64.1 ± 2.12 ^b^	46.4 ± 2.58 ^c^
Rolling (8–15)	66.2 ± 1.73 ^d^	55.9 ± 2.29 ^d^
Strong rolling (16–20)	69.6 ± 1.86 ^e^	62.0 ± 2.10 ^e^
Moderately steep (21–25)	67.2 ± 2.04 ^d^	41.9 ± 2.25 ^b^
Steep (26–35)	66.0 ± 2.35 ^c^	36.5 ± 2.34 ^a^
Very steep (35–75)	50.7 ± 2.91 ^a^	61.9 ± 3.05 ^e^

^a–e^ Within a treatment group (column), means with different letters are significantly different (*p* < 0.05). Restricted = Ewe access to the water trough was restricted. Unrestricted = Ewe access to water trough was not restricted.

## Data Availability

Not applicable.

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
