# Peer review of "The Behaviour of Sheep around a Natural Waterway and Impact on Water Quality during Winter in New Zealand"

_animals, 2023, doi:10.3390/ani13091461_

Round 1

Reviewer 1 Report

The article by Bunyaga et al. evaluated the behavior of sheep around a natural waterway and its potential interaction with water quality. The study is quite interesting and would be great values to sheep producers and the government agencies involved in managing waterways in New Zealand and other areas with similar sheep production systems. The study was well-crafted, the write-up was easy to follow, and the flow was quite engaging. Some minor suggestions and comments are provided below:

Line 22: Please concise the abstract to 250 words as per the journal guidelines.

Line 32-33: It would be helpful for the reader to easily understand the text if the information on GPS location data is mentioned first and then the comparison of access to water trough.

Line 46: Introduction section provided strong arguments to establish the need of the study. However, the fourth paragraph primarily justifies the use of GPS methods rather than contributing to the study's importance and limits the flow of the information. It may be reduced to 1-2 lines and further details could be provided in material and methods section.

Line 116: The description and the presentation of the study site in figures is excellent. The use of ArcGIS really produced high-quality figures.

Line 123-124: Would it be possible that the vegetation near waterway was of different type or quality/quantity than the other areas?

Overall, the manuscript has very well explained material and methods section.

The hot spots in figures 8C are not quite visible. May be a change of color pattern or marking would be of some help.

It would be great to add some information in the discussion section regarding the application and limitations of the study findings across different seasons.

Author Response

We thank the reviewer for identifying the weaknesses in our paper and providing us the opportunity to strengthen our research prior to publication. We found the comments extremely helpful and have revised accordingly. We have responded to them individually, indicating how we addressed each concern or problem and describing the changes we have made. We hope the revised manuscript will better suit the Journal of Animals.

General comments reviewer #1

Line 22: Please concise the abstract to 250 words as per the journal guidelines.

  • Accepted and the wording reduced to 250 words; addressed in lines 22 to 44.

Line 32-33: It would be helpful for the reader to easily understand the text if the information on GPS location data is mentioned first and then the comparison of access to water trough.

  • Accepted, however, the sentence containing this information was removed when condensing the abstract to 250 words.

Line 46: Introduction section provided strong arguments to establish the need of the study. However, the fourth paragraph primarily justifies the use of GPS methods rather than contributing to the study's importance and limits the flow of the information. It may be reduced to 1-2 lines and further details could be provided in material and methods section.

  • Accepted, the paragraph has been reduced to 2 lines as addressed in lines 75 to 84.

Line 116: The description and the presentation of the study site in figures is excellent. The use of ArcGIS really produced high-quality figures.

  • Thank you

Line 123-124: Would it be possible that the vegetation near waterway was of different type or quality/quantity than the other areas?

  • When determining the pasture mass and moisture, samples were randomly collected across the paddock including near the waterway. Moreover, one hundred readings were taken across the paddock at approximately 2m intervals and the average mass for the paddock recorded (as described in lines 225 to 227). We believe this method has provided a true representation of the whole paddock including the vegetation near the waterway.

Overall, the manuscript has very well explained material and methods section.

  • Thank you.

The hot spots in figures 8C are not quite visible. May be a change of color pattern or marking would be of some help.

  • Accepted and addressed in lines 414 and 415; a white arrow and word “HS” added on the map.

It would be great to add some information in the discussion section regarding the application and limitations of the study findings across different seasons.

  • Accepted, limitation has been added under discussion section as addressed in lines 537 and 596.

Reviewer 2 Report

I feel this manuscript is well written and is well put together.

I feel that the study design may be weak as it forces the animals to graze two weeks in the same pasture. One method to remedy this would have been to find a second watershed which was similar in size and shape but can be difficult to due. I would have split the plot into two plots and grazed the crossover design using restricted water and available trough as your two start points and allowing the crossover. 

It is difficult for research to use proprietary data from technology companies such as your accelerometer. Can you explain more of the data that came from them and how you used it within your analysis? What type of numbers were collected from the accelerometer and how were you able to use it in your results because it seems like you could have answered the behavior questions with only camera data.

Line 88 - I would remove i.e. 30% dry matter or put () around them

Line 80 - remove the comma after 30-32 

Author Response

We thank the reviewer for identifying the weaknesses in our paper and providing us the opportunity to strengthen our research prior to publication. We found the comments extremely helpful and have revised accordingly. We have responded to them individually, indicating how we addressed each concern or problem and describing the changes we have made. We hope the revised manuscript will better suit the Journal of Animals.

Comments reviewer #2

I feel this manuscript is well written and is well put together.

  • Thank you.

I feel that the study design may be weak as it forces the animals to graze two weeks in the same pasture. One method to remedy this would have been to find a second watershed which was similar in size and shape but can be difficult to due. I would have split the plot into two plots and grazed the crossover design using restricted water and available trough as your two start points and allowing the crossover.

  • We agree with this observation as we had the same thinking before the commencement of this study. However, no suitable other locations were identified on the study farm as they would need to be in another watershed. Moreover, both pasture mass and moisture were measured before the start of the study to be ensure that sheep had enough feed for the entire two weeks. It was not possible to divide the paddock due to the topography of the study site.

It is difficult for research to use proprietary data from technology companies such as your accelerometer. Can you explain more of the data that came from them and how you used it within your analysis? What type of numbers were collected from the accelerometer and how were you able to use it in your results because it seems like you could have answered the behaviour questions with only camera data.

  • The only accelerometer data used was the Bluetooth received signal strength indicator value (RSSI). This data was then used to calculate the distance between beacons and receivers using an equation determined from previous research. We could have answered most of the behaviour questions with only the camera data, but the accelerometers provided an estimate of the distance of animals from each camera location within the riparian zone. It also provided the number of sheep that were detected within 3m distance on either side of the stream.

Line 88 - I would remove i.e., 30% dry matter or put () around them

  • Accepted and addressed in line 90, “i.e., 30% dry matter” deleted.

Line 80 - remove the comma after 30-32 

  • Accepted and addressed in line 84; comma removed,

Reviewer 3 Report

Materials and methods were well described.

Statistical analyses were correctly defined and applied.

Results were given well order and clear.

This manuscript can be accepted after minor revision.

1. Lines 50-51:  Could you give ratio of animal and fertilizer originated N and P contaminations within total, it may be more meaningful.

2. Lines 66-67: What is contamination values of cattle, pigs and deer on waterways? And what is acceptable level or threshold value?

3. Line 105: “40.3346° S, 175.7316° E” here is not the exact place mentioned in Figure 1.

4. Line 234: It can be written as (L/s).

5. Line 324: It is only proposal that you can use “graphical line sample such as (-----)(…………..)” instead of “solid line”

6. Lines 331-332: “Correlations between flowrate and weather parameters showed a negative correlation between flowrate and daily average temperature (r2=0.2 p=0.002).” r2 is not correlation coefficient, don’t give its square, it leads to different statistic.

7. In Table 3. “Arithmetic mean” can be “Mean” only because the word Mean refers to arithmetic mean.

8. In Table 3. Please use P value instead of NS. Readers may want to see how far from the threshold.

9. Line 282: Please define which method used for which traits. For example for Figure 6, ANOVA is OK for Nitrate but Sediment, e-coli and Total phosphorus because of heteroscedasticity.

10. Line 416 and 439: “abcdef”, “ab” is enough. Is it for complete table? Please mention it clearly.

11. Lines 586-608: In this paragraph you can also mentioned the nitrogen leaching from soil to water. İt is known as effective.

12. In the Conclusion part: Talk about something on behavior according to the title.

Author Response

We thank the reviewer for identifying the weaknesses in our paper and providing us the opportunity to strengthen our research prior to publication. We found the comments extremely helpful and have revised accordingly. We have responded to them individually, indicating how we addressed each concern or problem and describing the changes we have made. We hope the revised manuscript will better suit the Journal of Animals.

Comments reviewer #3

Materials and methods were well described.

Statistical analyses were correctly defined and applied.

Results were given well order and clear.

This manuscript can be accepted after minor revision.

  • Thank you.

  1. Lines 50-51:  Could you give ratio of animal and fertilizer originated N and P contaminations within total, it may be more meaningful.
  • Accepted and addressed in lines 52 to 55; sentences “For example, fertilizer and dung as sources of P contribute up to 40% of the total and the rest is from other sources such as pasture-plants and soil components [7]” and “Fertiliser and manure account for 57% of the total nitrogen contamination in watersheds” added.

  1. Lines 66-67: What is contamination values of cattle, pigs and deer on waterways? And what is acceptable level or threshold value?
  • We do not have the information about the exact contamination values of cattle, pigs and dear on waterways. In New Zealand, concentrations of coli in agricultural streams are typically 20 times higher than streams in forested catchments (Davies-Colley et al., 2004). For example, of nitrate leached from livestock, 65% was from dairy, 26 from beef and 15% from sheep https://www.stats.govt.nz/indicators/nitrate-leaching-from-livestock

  • According to the National Policy Statement for Freshwater Management 2020, the current national Microbiological Guidelines for Freshwater Recreation Areas state that at coli concentrations should not be above 540 cfu/100 ml, the national bottom line for ammonia is 0.40mg/L, nitrate is 3.5mg/l and Phosphorus is 0.035mg/l (these values are the 95 percentiles). Addressed in lines 626, 627, 634 and 635.
  1. Line 105: “40.3346° S, 175.7316° E” here is not the exact place mentioned in Figure 1.
  • Accepted, “40.3346° S, 175.7316° E” represented the description of the Tuapaka farm in general and not the study paddock as in Fig 1. The paddock was about 615m away from the mentioned location with coordinates “40.3345° S, 175.7390° E”. Addressed in lines 107 and 108.
  1. Line 234: It can be written as (L/s).
  • Accepted and addressed in line 248; modified to read as (L/s)
  1. Line 324: It is only proposal that you can use “graphical line sample such as (-----)(…………..)” instead of “solid line”
  • Accepted and addressed in lines 338 and 339.
  1. Lines 331-332: “Correlations between flowrate and weather parameters showed a negative correlation between flowrate and daily average temperature (r2=0.2 p=0.002).” r2is not correlation coefficient, don’t give its square, it leads to different statistic.
  • Accepted and addressed in line 346; “r2=0.2” replaced with “r=0.4”
  1. In Table 3. “Arithmetic mean” can be “Mean” only because the word Mean refers to arithmetic mean.
  • Addressed in lines 363 to 364 “arithmetic” removed.
  1. In Table 3. Please use P value instead of NS. Readers may want to see how far from the threshold.
  • Accepted and addressed in lines 363 to 364; P-values added.
  1. Line 282: Please define which method used for which traits. For example for Figure 6, ANOVA is OK for Nitrate but Sediment, e-coli and Total phosphorus because of heteroscedasticity.
  • Thank you for this observation. To control heteroskedasticity, sediment, e-coli and Total phosphorus were transformed using Tukey's Ladder of Powers transformation. Non-parametric methods were used thereafter for data that was not normally distributed as mentioned in lines 309 and 310.
  1. Line 416 and 439: “abcdef”, “ab” is enough. Is it for complete table? Please mention it clearly.
  • Thank you for the observation. Means with different letters “abcdef” were significantly different within a treatment group (column). Addressed in lines 430 and 453 to make it clear; word “(column)” added.
  1. Lines 586-608: In this paragraph you can also mentioned the nitrogen leaching from soil to water. İt is known as effective.
  • Accepted, “from the soil” has been added in line 608.

  1. In the Conclusion part: Talk about something on behavior according to the title.
  • Accepted and addressed in line 701; word “Behaviour” added.

Reviewer 4 Report

Aim of the study is important and the behaviour observetions have scientific impact. Somtimes less is more - it would be better to focus on the behaviour of the animals, GPS and accelerometers' results and nothing else.

Recommendations and comments of the reviewer are listed in the manuscript attached. Most important ones are the followings:

Detailed information is needed about the background and management.

Results of the pasture measures are missing from the manuscript. It has an important role due to drinking behaviour and water consumption of the animals, as well.

Heat stress is mentioned several times in the text, while the study was done in winter.

Far-reaching conclusions have been made.

Author Response

We thank the reviewer for identifying the weaknesses in our paper and providing us the opportunity to strengthen our research prior to publication. We found the comments extremely helpful and have revised accordingly. We have responded to them individually, indicating how we addressed each concern or problem and describing the changes we have made. We hope the revised manuscript will better suit the Journal of Animals.

Comments reviewer #4

Aim of the study is important and the behaviour observations have scientific impact. Sometimes less is more - it would be better to focus on the behaviour of the animals, GPS and accelerometers' results and nothing else.

  • We feel that the water data is an important component of this paper, however, we recognise that in its current form the discussion of the data is quite long. We have revised the water quality discussion to be more succinct.

It is true that sometimes less is more, however, one of the key aims of the current study was to investigate whether behaviour of sheep around a natural stream would impact water quality in the stream as this is a critical question that needs to be examined in NZ. The current government stock exclusion policy does not include sheep, but this policy approach is based on anecdotal observations only, which assume that sheep prefer not to enter waterways. If NZ is to meet its water quality objectives, it is critical that we study and also publish the impact or otherwise of sheep on water quality. Many studies have previously done behavioural trials using digital technology but have not associated these with water quality and there is a paucity of information for sheep. The uniqueness of the study comes from combining, technology, behaviour and water quality data, which is of particular importance to NZ.

The mentioned regulation use the followings: "wetlands, lakes and rivers, and also "water bodies". It is recommended to use these words (e.g. water bodies) in the text.

  • We feel that waterway or stream is the most accurate way to describe the waterway examined in this study as ’waterbody’ is too generic and ‘river’ refers to a large body of flowing water, whereas the stream studied in this thesis, was less than 0.5m wide.

  1. Line 25: 1m or 1 m?
  • It is 1m, however, the whole sentence containing the word “1m” was removed when condensing the abstract to 250 words.

  1. Line 31: 3 m or 3m? This is a problem throughout the manuscript. Format, using and writing of same dimensions differ several times and are not uniform.
  • Accepted and addressed throughout the manuscript in lines 31, 416, 458, 569 and 580; “3 m” replaced with “3m”.

  1. Lines 117 to 125: Detailed information is needed about:

- breed of the animals (please repeat here what earlier has been mentioned);

- keeping system (supplementations if any; stocking in the paddock 0-24 hours or not; electric shepherd or fence has been used; stockperson or dog or nothing; SRG or MIRG and which No. of rotation has been used etc.);

  • Accepted and addressed in lines 112 and 121 to 128; information about breed, fencing, grazing method, feeding, and familiarity with the paddock added.

  1. Line 118: Have you used adaptation period? Were the animals grazed in this paddock before the study?
  • Addressed in line 125 to 126: No adaptation period employed as they were used to this paddock.

  1. Line 200: Where are the results of these measures?
  • Pasture mass and moisture content results were summarised and mentioned in lines 136 to 138.

  1. Line 202: were
  • Accepted and addressed in line 215; “was” replaced with “were”.

  1. Line 220: Previously (in Chapter 2.1.3.) you wrote: "The cameras also contained a built-in temperature sensor..." What was the role of this when you had a weather station?
  • Addressed in lines 157 to 158. It is true that cameras also contained a built-in temperature sensor, however, temperature data from the cameras were not used in this study. A sentence “The cameras also contained a built-in temperature sensor that recorded the ambient temperature” deleted.

  1. Line 505: Results?
  • Pasture mass and moisture content results were summarised and mentioned in lines 136 to 138.

  1. Line 509: What is the local climate classification?
  • New Zealand falls under temperate climate, however, tropics was used here as most of such studies were done in tropical, arid and semi-arid zones, Addressed in line 522; a sentence “There are limited studies in temperate climates such as NZ” added.

  1. Lines 511 and 502: I do not agree because of:

- Romney is an English breed, not a tropical one;

- there were no harsh circumstances (eg. heat stress or lack of precipitation) during this study which could force the animals to search water or pasture.

  • Accepted and addressed in line 526; in search of “water” removed and “lush” added, however, in search of pasture still holds due to sheep preference for lush pasture (Oluju, 2017). http://hh.diva-portal.org/smash/record.jsf?pid=diva2%3A1149919&dswid=-8865 . Also, sheep’s selective nature and their preference for green leaf (Lynch et al., 1992).

  1. Line 519 and 520: The study has been done during winter-time. It would be good to see the results of the same study in summer. The conclusions are far-reaching.
  • Thank you, we did another set of study in summer and the results were different from the current study, with sheep spending more time drinking from the stream in summer.
  1. Line 535: There was no heat stress, there was no summer, and water consumption and proportion of drinking behaviour was normal. Nothing should be concluded from this.
  • Accepted and addressed in lines 546 to 556; wording from literature in summer were removed and a sentence “It is likely that in the current study the ewes had satisfied their water needs through the consumption of pasture. Brown and Lynch [35] and McFarlane and Howard [36] suggest that if moisture content of the pasture is above 50% sheep do not need to drink free water [35,36].” added.

  1. Line 568 and 569: Air and water temperature should be also a key question. Maximum temperature during the study was 13.1 celsius. What was the temperature of the water?
  • Addressed in line 595. Unfortunately, water temperature was not measured and has been addressed as a limitation.

  1. Line 645 and 649: In winter!!!
  • Accepted and addressed in lines 696 and 702; word “in winter” added.

  1. Line 654: Totally agree!
  • Thank you.

Round 2

Reviewer 4 Report

Thank you very much for the revised version of the manuscript. It was improved and (almost) all of my recommendations have been accepted by the authors.